# OpenReview forum: "OFFSIDE: Benchmarking Unlearning Misinformation in Multimodal Large Language Models"
_ICLR.cc/2026/Conference — ICLR 2026 Conference Withdrawn Submission_

### Official Review · Reviewer_qQfk · 2025-10-21

**Soundness:** 2
**Presentation:** 3
**Contribution:** 3
**Rating:** 4
**Confidence:** 5

**Summary:**

This paper proposes a new benchmark, offside, to evaluate machine unlearning methods in multimodal large language models. OFFSIDE collects the image and information from real-world football players from google engines. This paper also evaluates several existing methods in unimodel unlearning to compare their performance.

**Strengths:**

1. This paper summarizes four settings of unlearning in MLLMs, which are significant contributions in the research field.

2. The rumors of images and texts in the football fiedl are well motivated to construct this benchmark,

3. The paper is well-written and easy to understand.

4.The unimodel unlearning methods are evaluated comprehensively across different tasks.

**Weaknesses:**

Despite a novel benchmark on MU in MLLMs, there are several drawbacks in this paper
1. As a benchmark paper, several multimodal methods are not evaluated such as [1] [2], which did not provide a comprehensive evaluation of unlearning methods.

2. Table 1 lists MMUBench, however, in the introduction section this benchmark is not metioned,  especially in line second paragraph, where related works are introduces.

3. The football field is rather limited in the real-world setting. Some other fields needs to be evaluated such as politicians and singers, etc.

4. I think classification and generation task are somewhat redundant. Classification is also a type of generation in the LLM tasks.




## References ##

[1] Huo, Jiahao, et al. "Mmunlearner: Reformulating multimodal machine unlearning in the era of multimodal large language models." arXiv preprint arXiv:2502.11051 (2025).

[2] Liu, Zheyuan, et al. "Modality-aware neuron pruning for unlearning in multimodal large language models." arXiv preprint arXiv:2502.15910 (2025).

**Questions:**

See weaknesses.

---

> ### Author Response · Authors · 2025-11-18
> **Reply to weaknesses**
>
> Dear Reviewer qQfk:
>
> Thank you for your suggestions! We have added additional experiments and explanations(highlighted in red) in the next version of our paper . We hope this will improve the clarity of our work.
>
> -----
> ***W1: As a benchmark paper, several multimodal methods are not evaluated such as [1] [2], which did not provide a comprehensive evaluation of unlearning methods.***
>
> A1: We have included the results of these methods in the next version of our paper. While their overall performance surpasses other baselines across all settings, they still exhibit the issues highlighted in our paper. This proves that our findings are broadly existed in existing methods.
>
> -----
>
> ***W2: Table 1 lists MMUBench, however, in the introduction section this benchmark is not mentioned, especially in line second paragraph, where related works are introduces.***
>
> A2: Thanks for your kind reminder! We have revised the related work section("MMUBench is tailored to the unlearning of real-world entities.").
>
> -----
> ***W3: The football field is rather limited in the real-world setting. Some other fields needs to be evaluated such as politicians and singers, etc.***
>
> A3: The contribution of OFFSIDE does not lie in the domain of the data. The core difference between OFFSIDE and other benchmarks lies in the visual rumors present in the images. While inspired by football rumors, this approach is not limited to this domain and can be generalized to other areas. For instance, we can use image-editing techniques to insert private information (text) into any singer's images, addressing the visual unlearning target problem. Additionally, we can easily mix benign information with harmful content related to a given entity (image) to simulate a selective unlearning scenario.
>
> -----
> ***W4: I think classification and generation task are somewhat redundant. Classification is also a type of generation in the LLM tasks.***
>
> A4: We follow the MLLMMU-Bench[1] setting here. Classification differs from generation tasks because the answer is embedded in the prompt for classification tasks. From this perspective, classification can be seen as a form of prompt attack, leading to significant differences in results. After unlearning, the classification accuracy drops slightly compared to the generation scores. This demonstrate that existing methods are vulnerable to prompt attacks.
>
> -----
>
> [1] Liu, Zheyuan, et al. "Protecting Privacy in Multimodal Large Language Models with MLLMU-Bench." Proceedings of the 2025 Conference of the Nations of the Americas Chapter of the Association for Computational Linguistics: Human Language Technologies (Volume 1: Long Papers). 2025.
>
> [2] Huo, Jiahao, et al. "Mmunlearner: Reformulating multimodal machine unlearning in the era of multimodal large language models." arXiv preprint arXiv:2502.11051 (2025).

---

> ### Author Response · Authors · 2025-11-26
>
> We hope this message finds you well. As the discussion period is drawing to a close (with about one week remaining), we wanted to reach out to see if you might have any further comments, questions, or concerns that we could help address. We sincerely appreciate the time and thought you’ve invested in reviewing our work, and we would be more than happy to provide any additional information that could support your evaluation.
>
> Thank you once again for your kind attention and valuable feedback.

---

### Official Review · Reviewer_4cFW · 2025-10-26

**Soundness:** 2
**Presentation:** 2
**Contribution:** 2
**Rating:** 4
**Confidence:** 3

**Summary:**

This paper introduces OFFSIDE, a new benchmark for evaluating machine unlearning in multimodal large language models (MLLMs).
The benchmark focuses on football transfer rumors as a misinformation scenario and defines four evaluation settings: Complete, Selective, Corrective, and Unimodal unlearning. These settings are designed to assess an MLLM's ability to selectively forget and relearn knowledge across both textual and visual modalities.

The authors conduct systematic experiments with several representative unlearning methods under consistent experimental conditions and report the following key findings:
(1) unimodal text-based methods fail on multimodal rumors;
(2) unlearning efficacy is largely determined by catastrophic forgetting;
(3) all methods struggle when misinformation is visually grounded;
(4) unlearned knowledge can be easily recovered through relearning; and
(5) all methods remain vulnerable to prompt-based attacks.

**Strengths:**

**S1. Systematic Formulation of Evaluation Settings**

This paper organizes four unlearning scenarios, i.e., Complete, Selective, Corrective, and Unimodal, within a single benchmark. Although several of these aspects have been individually explored before, presenting them together in a unified benchmark offers practical value for researchers studying multimodal unlearning.


**S2. Comprehensive Experimental Comparison**

This paper provides a broad empirical comparison of representative unlearning methods under consistent experimental conditions. Even if the findings largely align with prior studies, the inclusion of multiple modalities and evaluation aspects helps confirm and contextualize known challenges such as catastrophic forgetting and prompt vulnerability.

**Weaknesses:**

**W1. Scope of Benchmark**

The OFFSIDE benchmark focuses solely on football transfer rumors, which constitutes a narrow and specialized scenario. In contrast, other multimodal unlearning benchmarks such as MLLMU-Bench, CLEAR, and PEBench are designed around more general and socially relevant domains (e.g., person profiles, privacy, or harmful content) that better reflect practical application contexts for unlearning.

The paper does not adequately explain how misinformation about football transfers impacts the real world or what specific harms could be mitigated through unlearning in this domain. The only passage that briefly touches on this issue, referring to "demonstrating the practical value of multimodal unlearning in real-world applications" (L157), is superficial and lacks substantive discussion of social impact or practical relevance. Without such justification, it remains unclear whether the chosen domain can serve as a representative real-world use case for unlearning tasks.


**W2. Novelty of Benchmark**

The proposed "Corrective Relearning" setting appears to substantially overlap with knowledge editing, which likewise aims to modify or restore specific knowledge already learned by a pre-trained LLM/MLLM. Existing datasets and benchmarks such as [a-e] already evaluate this process by measuring knowledge correction accuracy, generalization, and side-effects after targeted knowledge updates. The authors need to clearly articulate conceptual or methodological distinctions from these knowledge editing benchmarks.

[a] Meng et al., Locating and Editing Factual Associations in GPT, ICLR 2022.

[b] Zhong et al., MQuAKE: Assessing Knowledge Editing in Language Models via Multi-Hop Questions, EMNLP 2023.

[c] Huang et al., VLKEB: A Large Vision-Language Model Knowledge Editing Benchmark, NeurIPS 2024.

[d] Zhang et al., MC-MKE: A Fine-Grained Multimodal Knowledge Editing Benchmark Emphasizing Modality Consistency, ACL findings 2025.

[e] Du et al., MMKE-Bench: A Multimodal Editing Benchmark for Diverse Visual Knowledge, ICLR 2025.


**W3. Composability of Existing Benchmarks**

**W3-1.** Given that existing benchmarks already cover unimodal, selective, and corrective (see W2) aspects separately, their combination could approximate the proposed benchmark. This raises questions about the necessity and standalone contribution of OFFSIDE as a new benchmark.

**W3-2.** The reported findings, namely, (1) unimodal methods fail on multimodal rumors, (2) unlearning efficacy is largely driven by catastrophic forgetting, (3) all methods struggle with visual rumors, (4) unlearned rumors can be easily recovered, and (5) all methods remain vulnerable to prompt attacks, are mostly consistent with observations reported in prior unimodal/multimodal unlearning studies (e.g., CLEAR, PEBench, MMUBench). It is unclear whether these insights uniquely emerge from OFFSIDE, or whether they simply reinforce existing knowledge observed across benchmarks.


**W4. Clarity of Dataset Construction Process**

Some important details about the dataset construction process are missing. In particular, the paper does not specify the criteria for selecting players, the criteria for labeling rumors vs. facts, or the reliability of data sources (e.g., official media outlets vs. fan forums).

**Questions:**

**W1.** Could the authors elaborate on why football transfer rumors were chosen as the domain for this benchmark? In particular, how does this setting reflect practical and socially relevant unlearning scenarios (e.g., personal data deletion, misinformation mitigation, or legal compliance)? Please also clarify any social impacts or safety motivations that justify the choice of this domain.

**W2.** How is the proposed Corrective Relearning setting conceptually or methodologically different from existing knowledge editing frameworks such as COUNTERFACT, MQuAKE, VLKEB, MC-MKE, or MMKE-Bench? Does the task introduce any new evaluation dimensions beyond what these benchmarks already measure?


**W3.** Several findings reported in the paper (e.g., catastrophic forgetting, recovery of unlearned information, and vulnerability to prompt attacks) seem consistent with prior unlearning studies (e.g., CLEAR, PEBench, MMUBench). Could the authors clarify which observations are unique to OFFSIDE and could not have been discovered through combinations of these existing benchmarks? Highlighting any domain-specific phenomena would strengthen the case for the necessity of OFFSIDE.


**W4.** Please provide more details about the dataset construction process. Specifically:

- What criteria were used to select the 80 players?

- How were rumors and facts labeled or verified (e.g., by which sources and at what time)?

- What are the primary data sources (official media outlets, news APIs, social media, fan forums, etc.), and how was their reliability ensured?

- Are there plans to release the dataset or documentation (e.g., annotation guidelines, license information) to support reproducibility?

---

> ### Author Response · Authors · 2025-11-18
> **Reply to w1**
>
> Dear reviewer 4cFW:
> We really appreciate your constructional suggestion! We will address each of your proposed weaknesses and questions one by one, providing detailed explanations and clarifications to ensure that all aspects of our work are thoroughly understood.
>
> ------
> ***W1. The OFFSIDE benchmark focuses solely on football transfer rumors, which constitutes a narrow and specialized scenario. In contrast, other multimodal unlearning benchmarks such as MLLMU-Bench, CLEAR, and PEBench are designed around more general and socially relevant domains (e.g., person profiles, privacy, or harmful content) that better reflect practical application contexts for unlearning.***
>
> A1:
> The contribution of OFFSIDE does not lie in the domain(football) of the data. The core difference between OFFSIDE and other benchmarks lies in the visual rumors present in the images. While inspired by football rumors, this approach is not limited to this domain and can be generalized to other areas. For instance, we can use image-editing techniques to insert private information (text) into images, addressing the visual unlearning target problem. Additionally, we can easily mix benign information with harmful content related to a given entity (image) to simulate a selective unlearning scenario. "These demonstrate that OFFSIDE is not confined to the football domain. On the contrary, it addresses a much broader, more general problem.
>
> We have updated the social impacts of football rumors in the Section 3.1(in red), highlighting the significance of OFFSIDE:
>
> Misinformation about football transfers can have significant real-world consequences. False rumors often lead to emotional reactions from fans, causing unnecessary excitement or disappointment. Unlearning techniques can mitigate these harms by preventing the spread of misinformation and ensuring decision-making is based on verified information. Moreover, the issue of football rumors is not isolated; it can generalize to other domains, such as sports journalism, social media, and financial markets, where rumors are prevalent. Unlearning such rumors is crucial for preserving trust, reducing instability, and promoting more reliable information across various societal sectors.
>
> ------

---

> ### Author Response · Authors · 2025-11-18
> **Reply to w2 and w3**
>
> ------
> ***W2. The proposed "Corrective Relearning" setting appears to substantially overlap with knowledge editing, which likewise aims to modify or restore specific knowledge already learned by a pre-trained LLM/MLLM. Existing datasets and benchmarks such as [a-e] already evaluate this process by measuring knowledge correction accuracy, generalization, and side-effects after targeted knowledge updates. The authors need to clearly articulate conceptual or methodological distinctions from these knowledge editing benchmarks.***
>
> [a] Meng et al., Locating and Editing Factual Associations in GPT, ICLR 2022.
>
> [b] Zhong et al., MQuAKE: Assessing Knowledge Editing in Language Models via Multi-Hop Questions, EMNLP 2023.
>
> [c] Huang et al., VLKEB: A Large Vision-Language Model Knowledge Editing Benchmark, NeurIPS 2024.
>
> [d] Zhang et al., MC-MKE: A Fine-Grained Multimodal Knowledge Editing Benchmark Emphasizing Modality Consistency, ACL findings 2025.
>
> [e] Du et al., MMKE-Bench: A Multimodal Editing Benchmark for Diverse Visual Knowledge, ICLR 2025.
>
> A2:
> We include the relearning setting primarily to enable a more comprehensive evaluation of the robustness of existing baselines. Specifically, we aim to ensure that unlearning methods can seamlessly integrate with other post-training techniques(e.g., model editing) to accomplish a complete downstream task. In the context of rumor mitigation, it is natural and indeed necessary to correct misinformation after selectively unlearning specific false claims. Therefore, we consider relearning an essential component of the pipeline. This is not the core contribution of OFFSIDE.
>
> Machine unlearning and model editing are two distinct research areas, each with its own data formats and evaluation standards. Model editing focuses on making targeted, precise modifications (preferably with an emphasis on locality) to a model’s behavior or knowledge, while machine unlearning aims to broadly remove specific information, prioritizing overall consistency. Currently, these two areas are typically studied in isolation. Due to their different objectives (e.g., target focus and data types), their evaluation methodologies also differ significantly, despite the fact that the evaluation metrics may be quite similar. Most importantly, both areas are still in the early stages within the context of MLLMs.
>
> OFFSIDE also offers valuable insights for model editing: How can we accurately edit rumors that appear in an image? How can we selectively edit the knowledge of a given image without damaging other benign information? And how can editing methods be scaled up to the unlearning domain? (Because, as we know, a prevailing paradigm in model editing is "locate-then-edit." However, as data scales up to more general unlearning tasks, a well-designed locate-then-edit method might encounter problems, such as over-editing, which could result in poor performance.)
>
> ------
>
> ***W3. Several findings reported in the paper (e.g., catastrophic forgetting, recovery of unlearned information, and vulnerability to prompt attacks) seem consistent with prior unlearning studies (e.g., CLEAR, PEBench, MMUBench). Could the authors clarify which observations are unique to OFFSIDE and could not have been discovered through combinations of these existing benchmarks? Highlighting any domain-specific phenomena would strengthen the case for the necessity of OFFSIDE.***
>
> A3:
> We present five key findings in OFFSIDE:
>
> (1) We find that unimodal unlearning methods fail to erase multimodal knowledge, which contrasts with conclusions drawn from MLLMMU-Bench[1](they observe effective unlearning using unimodal methods). We attribute this discrepancy to model collapse during unimodal unlearning observed in MLLMMU-Bench: rather than selectively forgetting the targeted content, these methods degrade the model’s overall capabilities, creating a false impression of successful unlearning.
>
> (2) Unlearning efficacy is largely driven by catastrophic forgetting. As far as we know, we are the first to conduct a quantitative experiment comparing the forgetting effects of unlearning methods with continual learning.
>
> (3) All methods struggle with "visual rumors" (rumors appearing in images); this is the core contribution of our paper which has not been addressed before.
>
> (4) The unlearned rumors can be easily recovered. While this has been explored in LLM MU[2], we aim to test whether this issue can be generalized to the MLLM context.
>
> (5) All methods are vulnerable to prompt attacks. While there have been works addressing prompt attacks in LLM MU[3,4,5], none have examined how these attacks affect classification tasks in MLLMs.
>
> These unique settings/findings strengthen the  necessity of OFFSIDE.

---

> ### Author Response · Authors · 2025-11-18
> **Reply to w4.**
>
> ------
>
> ***W4.
> Some important details about the dataset construction process are missing. In particular, the paper does not specify the criteria for selecting players, the criteria for labeling rumors vs. facts, or the reliability of data sources (e.g., official media outlets vs. fan forums).***
>
> A4: We have added the detail of the data construction in Appendix(line 920-929):
>
> The criteria for selecting the 80 players primarily depend on the ability to collect sufficient information, including rumor images and the corresponding rumors. This was a challenging task, as we reviewed nearly 200 players before identifying 80 players who met the requirements. All of the images were collected after the 2025 Premier League summer transfer window closed, when player information was relatively stable. The rumors were gathered from [Transfermarkt](https://www.transfermarkt.com/start). We hired two football experts to examine the images and corresponding texts twice to ensure their quality. Specifically, we first retrieved player information and associated transfer rumors from [Transfermarkt](https://www.transfermarkt.com/start). For the selected players, we then searched Google to find images corresponding to the text information (image-text association). Finally, we used GPT-4 to generate VQA pairs, which were used to construct the datasets.
>
> ------
> ***W4.1: Could the authors elaborate on why football transfer rumors were chosen as the domain for this benchmark? In particular, how does this setting reflect practical and socially relevant unlearning scenarios (e.g., personal data deletion, misinformation mitigation, or legal compliance)? Please also clarify any social impacts or safety motivations that justify the choice of this domain.***
>
> A4.1: The criteria mainly lie in whether we can collect enough information, including the rumor images and the corresponding misinformation. It is challenging and we browsed through nearly 200 players before finding 80 players who met the requirements.
>
> ------
> ***W4.2-3: How were rumors and facts labeled or verified (e.g., by which sources and at what time)? What are the primary data sources (official media outlets, news APIs, social media, fan forums, etc.), and how was their reliability ensured?***
>
> A4.2-3: All of the images were collected after the 2025 Premier League summer transfer window closed, when player information was relatively stable. All the rumors was gathered from ([https://www.transfermarkt.com/startseite](https://www.transfermarkt.com/startseite)). This is a website specifically designed for collecting transfermarket information. All of the images are collected from google research. We hired 2 football expert to examine the images and corresponding texts twice to ensure its quality.
>
> ------
> ***W4.4: Are there plans to release the dataset or documentation (e.g., annotation guidelines, license information) to support reproducibility?***
>
> A4.4: Sure! We will make it public available upon acceptance.
>
> ------
>
> [1] Liu, Zheyuan, et al. "Protecting Privacy in Multimodal Large Language Models with MLLMU-Bench." Proceedings of the 2025 Conference of the Nations of the Americas Chapter of the Association for Computational Linguistics: Human Language Technologies (Volume 1: Long Papers). 2025.
>
> [2] Yeonwoo Jang, et al. "Prompt Attacks Reveal Superficial Knowledge Removal in Unlearning Methods." COLM 2025 SoLaR Workshop.
>
> [3] Ziyao Liu, et al. "Threats, Attacks, and Defenses in Machine Unlearning: A Survey". arXiv, arxiv.org.
>
> [4] Vaidehi Patil, et al. "Unlearning Sensitive Information in Multimodal LLMs: Benchmark and Attack-Defense Evaluation" Published in Transactions on Machine Learning Research.
>
> [5]Yang Liu, et al. "Backdoor Defense with Machine Unlearning". INFOCOM.

---

> ### Author Response · Authors · 2025-11-26
>
> We hope this message finds you well. As the discussion period is drawing to a close (with about one week remaining), we wanted to reach out to see if you might have any further comments, questions, or concerns that we could help address. We sincerely appreciate the time and thought you’ve invested in reviewing our work, and we would be more than happy to provide any additional information that could support your evaluation.
>
> Thank you once again for your kind attention and valuable feedback.

---

> > ### Comment · Reviewer_4cFW · 2025-11-27
> >
> > Thanks to the authors for their efforts. The clarification regarding W4 (dataset construction process) is sufficiently addressed. However, I still think that W1-W3 remain unresolved.
> >
> > **W1. Scope of Benchmark**
> >
> > My main concern is that OFFSIDE is narrowly scoped, being constructed entirely around football rumors. This contrasts with existing multimodal unlearning benchmarks, which target broader and more socially consequential domains such as privacy, safety, or NSFW. I also note that Reviewer qQfk and Reviewer TYug independently raised the same concern, suggesting that this is not an isolated view.
> >
> > The rebuttal does not fully alleviate this concern:
> >
> > (i) For the societal impact of unleraning football rumors, the authors argue that fake football rumors may cause unnecessary excitement or disappointment among football fans. While I do not dispute that such effects matter, they are arguably less far, reaching compared with privacy-, safety-, and NSFW-related unlearning, where misinformation can have legal, ethical, or personal consequences for arbitrary individuals. In this sense, the potential scope of affected individuals remains comparatively limited.
> >
> > (ii) The rebuttal states that the benchmark is not about football but rather about visual rumors, and that the approach could generalize to other domains by inserting various types of information into images. However, the actual dataset is entirely confined to the football domain, and no empirical evidence is provided to demonstrate that this generalization is feasible and possible. As it stands, I feel that the generalizability of this benchmark beyond the football domain remains rather speculative.
> >
> >
> > **W2. Novelty of Benchmark**
> >
> > I acknowledge the value of jointly considering machine unlearning and knowledge editing (indeed some recent studies discussed them jointly [d]). However, the rebuttal still does not address my central question -- Is the proposed "corrective relearning" fundamentally different from knowledge editing? More concretely, the paper needs to clarify:
> >
> > (i) What conceptual or methodological distinction exists between corrective relearning and knowledge editing?
> >
> > (ii) How does the evaluation of corrective relearning differ from established knowledge editing evaluations?
> >
> > The rebuttal states that corrective relearning is not the core contribution of OFFSIDE. However, Table 1 explicitly lists corrective relearning as a differentiating component of the benchmark, which suggests that the authors do intend it to contribute to the benchmark's novelty. This makes a direct and explicit discussion even more necessary. As the rebuttal does not provide this clarification, the conceptual contribution of corrective relearning remains ambiguous.
> >
> >
> > **W3. Composability of Existing Benchmarks**
> >
> > The rebuttal argues that the five key insights represent novel findings revealed by OFFSIDE. However, I think that four out of the five have been fully or partially reported in the literature or can be evaluated using existing benchmarks. More specifically:
> >
> > (1) Failure of unimodal unleraning on multimodal unlearning has already been pointed out in prior multimodal unlearning work (e.g., [a,b]).
> >
> > (2) The relation between catastrophic forgetting and unlearning is widely discussed in the literature, e.g., [c].
> >
> > (4) Recovery of unlearned knowledge has been repeatedly observed in prior work, and robust unlearning has been identified as an open research challenge [d–f], even for MLLMs [g]. Since rumors are also a form of knowledge, the same phenomenon is expected to occur.
> >
> > (5) While I am not aware of any prior work that directly demonstrates vulnerability to prompt attacks in unlearning for MLLM-based classification, it is well established in the broader unlearning literature that many unlearning methods are highly susceptible to prompt attacks (e.g., [e,g]).
> >
> > As a result, I am still not convinced that these findings substantively establish the uniqueness or necessity of OFFSIDE as a standalone benchmark.
> >
> > [a] Liu et al., Protecting Privacy in Multimodal Large Language Models with MLLMU-Bench, NAACL 2025.
> >
> > [b] Cheng et al., MultiDelete for Multimodal Machine Unlearning, ECCV 2024.
> >
> > [c] Wang et al., A Comprehensive Survey of Forgetting in Deep Learning Beyond Continual Learning, PAMI 2025.
> >
> > [d] Guo et al., Mechanistic Unlearning: Robust Knowledge Unlearning and Editing via Mechanistic Localization, ICML 2025.
> >
> > [e] Yuan et al., Towards Robust Knowledge Unlearning: An Adversarial Framework for Assessing and Improving Unlearning Robustness in Large Language Models, AAAI 2025.
> >
> > [f] Zhang et al., Catastrophic Failure of LLM Unlearning via Quantization, ICLR 2025.
> >
> > [g] Ma et al., Benchmarking Vision Language Model Unlearning via Fictitious Facial Identity Dataset, ICLR 2025.

---

> > > ### Author Response · Authors · 2025-11-27
> > > **reply to w1 and w2**
> > >
> > > Thanks for your suggestions!
> > >
> > > W1. (i) For the societal impact of unleraning football rumors, the authors argue that fake football rumors may cause unnecessary excitement or disappointment among football fans. While I do not dispute that such effects matter, they are arguably less far, reaching compared with privacy-, safety-, and NSFW-related unlearning, where misinformation can have legal, ethical, or personal consequences for arbitrary individuals. In this sense, the potential scope of affected individuals remains comparatively limited.
> > >
> > > A: In the first round of rebuttal I misunderstood your point. In fact, the constructed data already covers privacy- and safety- unlearning. For each player, OFFSIDE provides two types of information (i.e., private and shared information), as illustrated in Figure 2. Private information is specifically designed for privacy-related unlearning (in the selective unlearning setting), while shared information is introduced to ensure that the unlearning process remains safe(i.e., it does not impair the model’s general knowledge). Moreover, fake football rumors primarily violate a player’s right to reputation and privacy, and can further harm their personality rights and legitimate economic interests linked to their public image and contracts. These issues may also lead to legal and ethical concerns. We will clarify these aspects more explicitly in the main paper. Thank you again for your helpful suggestions.
> > >
> > > (ii) The rebuttal states that the benchmark is not about football but rather about visual rumors, and that the approach could generalize to other domains by inserting various types of information into images. However, the actual dataset is entirely confined to the football domain, and no empirical evidence is provided to demonstrate that this generalization is feasible and possible. As it stands, I feel that the generalizability of this benchmark beyond the football domain remains rather speculative.
> > >
> > > A: Real-world data in which rumors explicitly appear in the image are extremely hard to obtain, and manual collection is both time-consuming and expensive, randomly synthesizing such rumors would risk infringing players’ privacy, reputation, personality rights, and even image- and contract-related economic interests. For this reason, we deliberately restrict our benchmark to rumors in the football domain. Moreover, since MLLM unlearning is still at an early exploratory stage, we believe the primary goal of a benchmark should be to demonstrate the potential practical value of MLLM-based unlearning, rather than to maximize domain coverage. In fact, domain breadth is not typically treated as the main criterion for a high-quality unlearning benchmark. For example, the widely used Harry Potter dataset [h] for LLM unlearning focuses on the content of a single fictional book, yet it is still broadly recognized as a standard and impactful dataset in this area.
> > >
> > > W2. (i) What conceptual or methodological distinction exists between corrective relearning and knowledge editing?
> > >
> > > A: Conceptually, corrective relearning and model editing share a similar high-level goal: both aim to transform harmful knowledge into correct and harmless behavior. Methodologically, however, corrective relearning is explicitly designed as a two-stage procedure, where the model first unlearns the problematic knowledge and then relearns the correct information. In contrast, existing editing approaches typically rely on a single post-training step, and to the best of our knowledge there are no editing methods that follow a “first unlearn, then relearn” paradigm as we do. In addition, the data formulations are quite different: corrective relearning is aligned with standard fine-tuning formats, whereas editing methods usually operate on triplet-style data (e.g., ⟨prompt, target output, locality constraints⟩). We overlooked this point previously, and we will add the above discussion to the Appendix to clarify the differences between corrective relearning and knowledge editing.
> > >
> > > (ii) How does the evaluation of corrective relearning differ from established knowledge editing evaluations?
> > >
> > > A: For Table 1, although there are some conceptual similarities, our selective unlearning setting is in fact different from PEBench; please see Reviewer TYug, comment W4 for a detailed discussion. Therefore, corrective relearning is not the only novelty at the task-setting level. The original motivation for introducing corrective relearning is to evaluate the robustness of unlearning methods: specifically, whether the forgotten rumors remain forgotten under a relearning attack. In practice, corrective relearning is better viewed as an engineering-style module that simulates how various real-world post-training procedures may be combined. Within this framework, our goal is to integrate more existing knowledge and perspectives so as to provide a more comprehensive and multidimensional evaluation of unlearning methods.

---

> > > ### Author Response · Authors · 2025-11-27
> > > **reply to w3**
> > >
> > > W3. Composability of Existing Benchmarks
> > >
> > > (1) Failure of unimodal unleraning on multimodal unlearning has already been pointed out in prior multimodal unlearning work (e.g., [a,b]).
> > >
> > > A: The main finding reported in MLLMMU-Bench [a] is that unimodal unlearning is slightly less effective than multimodal unlearning. In contrast, OFFSIDE shows that unimodal unlearning almost completely fails to erase the targeted rumors. In our experiments, we further observe that the models produced in [a] tend to suffer from overfitting or even model collapse, which we believe may partially obscure the true gap between unimodal and multimodal unlearning and thus lead to a somewhat misleading conclusion.
> > >
> > >
> > > (2) The relation between catastrophic forgetting and unlearning is widely discussed in the literature, e.g., [c].
> > >
> > > A: Because of the differences between multimodal large models and pure text-based large models, we cannot simply assume that findings about unimodal unlearning will generalize to the multimodal setting. As far as we know, no existing MLLM unlearning benchmark includes this specific setting or reports such findings. We acknowledge that our inclusion of this setting is primarily for the sake of a more comprehensive evaluation.
> > >
> > >
> > > (4) Recovery of unlearned knowledge has been repeatedly observed in prior work, and robust unlearning has been identified as an open research challenge [d–f], even for MLLMs [g]. Since rumors are also a form of knowledge, the same phenomenon is expected to occur.
> > >
> > > A: Because of the differences between multimodal large models and pure text-based large models, we cannot assume that findings about Recovery of unlearned knowledge will directly generalize to the multimodal setting. To the best of our knowledge, no existing MLLM unlearning benchmark includes this setting or reports such a finding. We therefore regard this setting primarily as a component for more comprehensive evaluation. In particular, the attacking methods used in [g] are not relearning attacks; they focus on adversarial prompting rather than retraining the model on the forgotten knowledge, so the underlying problem setup is essentially different from ours.
> > >
> > >
> > > (5) While I am not aware of any prior work that directly demonstrates vulnerability to prompt attacks in unlearning for MLLM-based classification, it is well established in the broader unlearning literature that many unlearning methods are highly susceptible to prompt attacks (e.g., [e,g]).
> > >
> > > A: This is largely because most existing benchmarks for MLLM unlearning do not include classification tasks, and even when they do [a], they do not explicitly analyze vulnerability to prompt-based attacks in this setting. To the best of our knowledge, OFFSIDE is the first to systematically highlight this issue for MLLM-based classification, and we believe it represents an important gap that future unlearning benchmarks and methods should address. Moreover, prompt attacks can take many different forms, such as paraphrasing-based attacks [e] or Dynamic Unlearning Attack [g], which optimize adversarial suffixes. These are conceptually different from the classification-style prompt attack we study, where the correct answer is directly embedded into the prompt to bypass the unlearning effect.
> > >
> > > In conclusion, we will revise the main paper by reorganizing the five findings into three primary contributions (corresponding to Findings 1, 3, and 5) and two domain-extension findings（2 and 4）, so that readers can more intuitively understand the contributions of OFFSIDE.
> > >
> > > [a] Liu et al., Protecting Privacy in Multimodal Large Language Models with MLLMU-Bench, NAACL 2025.
> > >
> > > [b] Cheng et al., MultiDelete for Multimodal Machine Unlearning, ECCV 2024.
> > >
> > > [c] Wang et al., A Comprehensive Survey of Forgetting in Deep Learning Beyond Continual Learning, PAMI 2025.
> > >
> > > [d] Guo et al., Mechanistic Unlearning: Robust Knowledge Unlearning and Editing via Mechanistic Localization, ICML 2025.
> > >
> > > [e] Yuan et al., Towards Robust Knowledge Unlearning: An Adversarial Framework for Assessing and Improving Unlearning Robustness in Large Language Models, AAAI 2025.
> > >
> > > [f] Zhang et al., Catastrophic Failure of LLM Unlearning via Quantization, ICLR 2025.
> > >
> > > [g] Ma et al., Benchmarking Vision Language Model Unlearning via Fictitious Facial Identity Dataset, ICLR 2025.
> > >
> > > [h] Ronen Eldan and Mark Russinovich. Who’s Harry Potter? Approximate Unlearning in LLMs. arXiv:2310.02238, 2023.

---

### Official Review · Reviewer_wAqb · 2025-10-30

**Soundness:** 3
**Presentation:** 3
**Contribution:** 3
**Rating:** 4
**Confidence:** 4

**Summary:**

This paper introduces OFFSIDE, a new benchmark for evaluating misinformation unlearning in Multimodal Large Language Models (MLLMs).  OFFSIDE is based on football transfer rumors and includes 15.68K records for 80 players. It provides four test sets to assess forgetting efficacy, generalization, utility, and robustness, and supports selective unlearning, corrective relearning, and unimodal unlearning. Experiments reveal key findings in current unlearning methods in each setting.

**Strengths:**

-  S1. The paper systematically divides the unlearning problem into Complete Unlearning, Selective Unlearning, Corrective Relearning, and Unimodal Unlearning, which facilitates a more fine-grained analysis.

-  S2. One of the main challenges in this field is the lack of datasets with a sufficient number of samples and the fact that most existing datasets are limited to facial images. In this regard, creating a dataset with a sufficient number of samples is highly commendable.

**Weaknesses:**

- W1. The interpretation of the results is somewhat unclear. For example, in L382-383, the paper states, “We observe that all the baselines exhibit a performance drop (compared to the vanilla model) in both private information and shared information.”. However, looking at Table 3, the Shared Information scores do not appear to decrease significantly for GD and KL. The Retain Set in Complete Unlearning also shows a similar drop. Therefore, the difference between the findings of Selective and Complete Unlearning seems minimal.

- W2. The claim made in Lines 379–381 is also confusing. The paper mentions overfitting to the Retain Set, yet the models achieve high scores on MMBench, which would not be consistent with overfitting. This shows that the baselines may have been underestimated.

- W3. The experiments are limited to LoRA fine-tuning. Since the knowledge acquired through LoRA fine-tuning is stored in a relatively small number of parameters, the results could differ from those obtained through full fine-tuning. Given that the Qwen model used is relatively small, including results with full fine-tuning and analyzing how the outcomes differ would make this paper more valuable.

- W4. The paper could further explore practical failure cases or qualitative analyses. Although it identifies the problems of existing methods in each unlearning category, it does not provide enough insight into how unlearning can be improved or concrete examples of failure cases. Including such an analysis or more failure cases would be particularly useful for a benchmark-focused paper.

**Questions:**

- I would like a more detailed explanation of the differences between the findings of Selective and Complete Unlearning.

- Regarding Lines 379–381, I would like to understand why the model is overfitting to the Retain Set, even though its performance on MMBench remains high.

- I am also interested in seeing the results of full fine-tuning.

- I would like to know if there are any insights for improving unlearning performance based on their analysis.

---

> ### Author Response · Authors · 2025-11-18
> **Reply to weaknesses**
>
> Dear reviewer wAqb:
>
> Thank you for your questions! We appreciate your thoughtful feedback and address each point in detail to clarify any concerns and improve our work.
>
> ------
> ***W1. The interpretation of the results is somewhat unclear. For example, in L382-383, the paper states, “We observe that all the baselines exhibit a performance drop (compared to the vanilla model) in both private information and shared information.”. However, looking at Table 3, the Shared Information scores do not appear to decrease significantly for GD and KL. The Retain Set in Complete Unlearning also shows a similar drop. Therefore, the difference between the findings of Selective and Complete Unlearning seems minimal.***
>
> A1. In the context of unlearning, it is reasonable that GD and KL achieve higher Shared Information scores in Selective and Complete Unlearning because they explicitly incorporate the Retain Set(Shared Information in Selective Unlearning) during unlearning (e.g., GD performs gradient ascent on the forget set while simultaneously performing gradient descent on the retain set). We have provided the introduction of these methods in the Appendix.
>
> ------
> ***W2. The claim made in Lines 379–381 is also confusing. The paper mentions overfitting to the Retain Set, yet the models achieve high scores on MMBench, which would not be consistent with overfitting. This shows that the baselines may have been underestimated.***
>
> A2. This observation stems from our empirical experiments. The fine-tuning step has a significant impact on the results: excessive fine-tuning risks overfitting. In fact, the results reported in MLLMU-Bench[1] appear to come from an overfitted model that has already lost its generalization capability. Comparing methods after model collapse is inherently unfair and misleading. To ensure a fair and meaningful comparison, we evaluate all methods on MM-Bench, guaranteeing that none of our reported models have collapsed. Therefore, the inclusion of MM-Bench serves as **a guarantee that all results represent a fair comparison.** In addition, different baselines employ different fine-tuning steps. This variability is difficult to control, and to the best of our knowledge, no prior work has systematically addressed this issue. In our evaluation, we explicitly avoid comparing results from collapsed or overfitted models. Instead, during the unlearning process, we record model performance every 10 training steps and report the best result before overfitting for existing methods. This approach ensures that our comparisons reflect genuine unlearning effectiveness rather than artifacts of degraded model capacity.
>
> ------
> ***W3. The experiments are limited to LoRA fine-tuning. Since the knowledge acquired through LoRA fine-tuning is stored in a relatively small number of parameters, the results could differ from those obtained through full fine-tuning. Given that the Qwen model used is relatively small, including results with full fine-tuning and analyzing how the outcomes differ would make this paper more valuable.***
>
> A3. Our experimental settings mainly follow previous works [1,2,3], all of which use LoRA fine-tuning. The reason for using LoRA lies in its efficiency, which aligns with the goal of machine unlearning: achieving unlearning of the Forget Set with minimal modification to the vanilla model (since LoRA fine-tuning only updates a subset of the parameters).
>
> ------
> ***W4. The paper could further explore practical failure cases or qualitative analyses. Although it identifies the problems of existing methods in each unlearning category, it does not provide enough insight into how unlearning can be improved or concrete examples of failure cases. Including such an analysis or more failure cases would be particularly useful for a benchmark-focused paper.***
>
> A4. We have provided representative failure cases in Figures 4 and 5. These failures are not coincidental.
>
> For example, as noted in the statement “All methods struggle with visual rumors (i.e., rumors embedded directly in the image),” we examined all such visual rumor instances in our benchmark and found that none of them were successfully unlearned by any method. We have revised our paper in line 454("We examined all instances of visual rumors in our benchmark and found that none were successfully unlearned by any method.").
>
> Regarding the conclusion that “machine unlearning largely stems from catastrophic forgetting,” we support this claim with qualitative analysis presented in line 473("Specifically, the unlearned sample IDs through GA, GD, KL, and NPO show 71\%, 48\%, 58\%, and 60\% similarity to the forgotten IDs after a simple relearning step.") of the paper. For "how unlearning can be improved", please refer to the reply to Q4.

---

> ### Author Response · Authors · 2025-11-18
> **Reply to questions**
>
> ------
> ***Q1: I would like a more detailed explanation of the differences between the findings of Selective and Complete Unlearning.***
>
> A1: Good question! The concept of selective unlearning in OFFSIDE is meaningfully different from prior work(Complete Unlearning). As stated in lines 80–81 of our paper: “The Selective Unlearning setting evaluates the ability to accurately erase specific image-text associations without affecting other learned knowledge.” In contrast, existing benchmarks [1,2] define unlearning as the removal of all private information associated with a given entity which we call it Complete Unlearning. However, OFFSIDE **focuses on a more fine-grained scenario**: for a single image that may be linked to multiple pieces of private information (e.g., 10 distinct facts), our benchmark requires unlearning only a subset of those associations (e.g., 4 out of 10 facts), while preserving the rest. This distinction is crucial because the retained information may be benign or publicly acceptable, and indiscriminately deleting all associated data would be unnecessarily destructive. Our setting, therefore, better reflects real-world requirements  rather than blanket removal, which is essential for privacy-preserving unlearning.
>
> ------
> ***Q2: Regarding Lines 379–381, I would like to understand why the model is overfitting to the Retain Set, even though its performance on MMBench remains high.***
>
> A2: Please see W2.
>
> ------
> ***Q3: I am also interested in seeing the results of full fine-tuning.***
>
> A3: Please see W3.
>
> ------
> ***Q4: I would like to know if there are any insights for improving unlearning performance based on their analysis.***
>
> A4: In OFFSIDE, we observe that “unlearned rumors can be easily recovered.” This raises critical questions: How exactly does the model perform unlearning? Why can seemingly forgotten knowledge be restored with simple attacks? To address these, future work could leverage interpretability tools such as neuron activation patterns or attention attribution to probe the internal mechanisms of unlearning in multimodal models. Moreover, we find that unimodal unlearning methods fail to erase multimodal knowledge, which contrasts with conclusions drawn from benchmarks like MLLMMU-Bench[1]. We attribute this discrepancy to model collapse during unimodal unlearning observed in MLLMMU-Bench: rather than selectively forgetting targeted content, these methods degrade the model’s general capabilities, creating a false impression of successful unlearning. This failure reveals a deeper issue: current unlearning approaches are still largely grounded in next-token prediction paradigms and exhibit strong modality bias. Knowledge across modalities is not jointly represented or edited, suggesting that effective multimodal unlearning requires a better understanding of how cross-modal knowledge is stored and entangled in MLLMs. We have added a section of future work in Appendix(in red).
>
> ------
> [1] Liu, Zheyuan, et al. "Protecting Privacy in Multimodal Large Language Models with MLLMU-Bench." Proceedings of the 2025 Conference of the Nations of the Americas Chapter of the Association for Computational Linguistics: Human Language Technologies (Volume 1: Long Papers). 2025.
> [2] AlexeyDontsov, et al. Clear: Character unlearning in textual and visual modalities. arXiv preprint arXiv:2410.18057,2024.
> [3] Huo, Jiahao, et al. "Mmunlearner: Reformulating multimodal machine unlearning in the era of multimodal large language models." arXiv preprint arXiv:2502.11051 (2025).

---

> ### Author Response · Authors · 2025-11-26
>
> We hope this message finds you well. As the discussion period is drawing to a close (with about one week remaining), we wanted to reach out to see if you might have any further comments, questions, or concerns that we could help address. We sincerely appreciate the time and thought you’ve invested in reviewing our work, and we would be more than happy to provide any additional information that could support your evaluation.
>
> Thank you once again for your kind attention and valuable feedback.

---

> > ### Comment · Reviewer_wAqb · 2025-11-28
> > **Official Comment by Reviewer wAqb**
> >
> > I appreciate the authors’ rebuttal. My concern regarding W3. LoRA Tuning has been resolved.
> > I would like to revisit a few remaining concerns as follows:
> >
> > W1.
> > My question was that the empirical findings for Complete Unlearning and Selective Unlearning appear very similar, since both settings exhibit comparable performance-drop patterns. However, the rebuttal primarily explained why GD and KL achieve higher Shared Information scores, rather than addressing the core point of my question. As a result, it still seems natural to conclude that the findings across the two settings do not meaningfully differ.
> > If that is the case, then the necessity of dividing the task into Complete Unlearning and Selective
> > Unlearning and evaluating them separately also becomes less convincing.
> >
> >
> > W2.
> > I am unsure about the claim that the models are overfitting to the Retain Set while they achieve high performance on MMBench (Lines 379–381).
> > However, in the rebuttal, the authors mainly explain why they adopt MMBench for evaluation, which does not answer my question.
> > As a result,  the statement of overfitting suggests that the baselines may have been underestimated.
> >
> > W4.
> > Although the failure cases primarily focus on visual rumors, it remains unclear how many visual rumors are actually included in this benchmark. Moreover, the benchmark does not sufficiently evaluate performance on non-visual-rumor cases, nor does it provide a deeper analysis of why these failures occur. Therefore, the feedback from the evaluation with this benchmark is inherently limited.
> >
> > I would appreciate it if the authors could address these remaining concerns.

---

> ### Author Response · Authors · 2025-11-28
> **reply to w1, w2 and w4**
>
> Thanks for your questions, we will reply them point by point.
>
> W1. My question was that the empirical findings for Complete Unlearning and Selective Unlearning appear very similar, since both settings exhibit comparable performance-drop patterns. However, the rebuttal primarily explained why GD and KL achieve higher Shared Information scores, rather than addressing the core point of my question. As a result, it still seems natural to conclude that the findings across the two settings do not meaningfully differ. If that is the case, then the necessity of dividing the task into Complete Unlearning and Selective Unlearning and evaluating them separately also becomes less convincing.
>
> A:The empirical results for **Complete Unlearning** and **Selective Unlearning** are very similar. This suggests that the evaluated unlearning baselines fail to selectively forget rumor information about a given image while preserving other benign knowledge. Instead, they appear to follow a **destructive shortcut**: achieving unlearning effects by erasing the entire identity of the player, rather than precisely unlearning the specific information associated with that individual. This finding highlights a fundamental limitation of existing unlearning methods in the multimodal setting, revealing that they struggle to perform fine-grained, information-level unlearning without resorting to destructive removal of entire identities.
>
>
> W2. I am unsure about the claim that the models are overfitting to the Retain Set while they achieve high performance on MMBench (Lines 379–381). However, in the rebuttal, the authors mainly explain why they adopt MMBench for evaluation, which does not answer my question. As a result, the statement of overfitting suggests that the baselines may have been underestimated.
>
> A: In the paper, we claim that “however, they are **prone** to overfitting to the D_relearn in practice. As a result, these methods require very careful control of the training process, which limits their practicality.” Our view is that the reported results must still achieve high performance on MMBench to ensure a fair comparison, which in turn indicates that the final models used for evaluation are not overfitting. However, similar to the model collapse that can be caused by gradient ascent, methods such as GD and KL that optimize on the Retain Set may easily lead to overfitting to the Retain Set, because it is difficult to precisely control the fine-tuning step. If the model is fine-tuned too much on the Retain Set, there is a clear risk of overfitting. This observation comes from our own reproduction of MLLMMU-Bench [a]. Fortunately, such overfitting and collapse issues can be detected by evaluating the model on MMBench.
>
> We hope this explanation clarifies what we mean by “prone to overfitting” in the paper.
>
> W4. Although the failure cases primarily focus on visual rumors, it remains unclear how many visual rumors are actually included in this benchmark. Moreover, the benchmark does not sufficiently evaluate performance on non-visual-rumor cases, nor does it provide a deeper analysis of why these failures occur. Therefore, the feedback from the evaluation with this benchmark is inherently limited.
>
> A: Thanks for your kind reminder! Because visual rumors are extremely difficult to collect, there is only **one** visual rumor for each image (and 8 rumors in total per player). We have 80 players, and each player has 8 images, resulting in **640 visual rumors** in total. Consequently, the dataset contains a mixture of visual-rumor and non-visual-rumor cases. Since prior work [a] has already focused on the non-visual-rumor setting, we place greater emphasis on the visual-rumor cases, which have not yet been systematically addressed. We will include these details in the main paper to make the setting clearer.
>
> Actually, we already provide an intuitive explanation for the failure of visual rumors in lines 456–459:  "This is intuitive because, even if the model forgets the visual rumors at the visual-text fusion level, it still lacks the necessary knowledge to correctly answer the question. As a result, the model's response primarily depends on the information it perceives in the image, without recognizing that the visual information is unreliable."
>
> I hope our reply could help you better understanding OFFSIDE.
>
> [a] Liu et al., Protecting Privacy in Multimodal Large Language Models with MLLMU-Bench, NAACL 2025.

---

### Official Review · Reviewer_TYug · 2025-11-03

**Soundness:** 3
**Presentation:** 2
**Contribution:** 2
**Rating:** 2
**Confidence:** 4

**Summary:**

Exisiting benchmarks do have selective forgetting
Domain is very niche

The paper introdices OFFSIDE a new benchmark for evaluating machine unlearning in multimodal large language models (MLLMs), focusing on misinformation such as football transfer rumors.
The dataset contains real-world vision–question–answer pairs across four evaluation settings: complete, selective, corrective relearning, and unimodal unlearning.
It enables studying how well models can selectively forget false or private information while retaining general knowledge.
Experiments with five unlearning baselines show that text-only (unimodal) methods fail on multimodal misinformation, and that most apparent unlearning results from catastrophic forgetting rather than targeted removal.
Overall, OFFSIDE reveals that current unlearning approaches remain ineffective, easily reversible, and vulnerable to prompt-based attacks.

**Strengths:**

- OFFSIDE introduces a real-world, benchmark based on football transfer rumors, representing a realistic domain where misinformation and visual evidence interact.
- The inclusion of four complementary unlearning scenarios (complete, selective, corrective relearning, and unimodal) adds diverse evaluation sets, extending prior unlearning paradigms to more complex multimodal and continual learning contexts.

**Weaknesses:**

- Certain sections (e.g., Table 1 and parts of the Introduction) are somewhat dense, making it harder for readers to immediately grasp how OFFSIDE concretely differs from prior datasets.

- While the paper introduces the “Corrective Relearning” scenario and lists four datasets (Forget Set, Retain Set, Test Set, Relearn Set), it only briefly mentions how these relate to continual learning (“simulates a continual learning framework to examine whether previously unlearned rumors can be successfully recovered after post-training”). However, it does not clearly explain:

- How examples in the Relearn Set are derived from the Forget Set (e.g., are they corrected versions of the same rumors, or new data about the same entities?).

- Whether the Retain Set is revisited or frozen during relearning, and how the model’s stability on it is assessed.

- The temporal or procedural link between unlearning and relearning phases — i.e., whether the model is incrementally trained on new data or simply fine-tuned again.

- The paper has limited novelty. Selective unlearning has been explored in previous multimodal unlearning datasets [1,2]

- Continual learning as defined is the paper is essentially the finetuning attack against unlearning and safety methods that previous works have explored [3, 1].

- The domain of the dataset is very niche and the findings may or may not generalize across domains.

[1] Patil, Vaidehi, et al. "Unlearning Sensitive Information in Multimodal LLMs: Benchmark and Attack-Defense Evaluation." Transactions on Machine Learning Research.

[2] Liu, Zheyuan, et al. "Protecting Privacy in Multimodal Large Language Models with MLLMU-Bench." Proceedings of the 2025 Conference of the Nations of the Americas Chapter of the Association for Computational Linguistics: Human Language Technologies (Volume 1: Long Papers). 2025.

[3] Qi, Xiangyu, et al. "Safety Alignment Should be Made More Than Just a Few Tokens Deep." The Thirteenth International Conference on Learning Representations.

**Questions:**

- Why does the continual learning set be a part of the dataset? It can be any other dataset, right? Is there a reason for the way it is designed?
- Is there a difference between the continual learning setting proposed in the paper and finetuning attacks proposed in previous papers?

---

> ### Author Response · Authors · 2025-11-18
> **Reply to weaknesses**
>
> Dear Reviewer TYug:
>
> Thanks for your suggestions! We will address each of your proposed weaknesses and questions one by one, providing detailed explanations and clarifications to ensure that all aspects of our work are thoroughly understood.
>
> ------
> ***W1: Certain sections (e.g., Table 1 and parts of the Introduction) are somewhat dense, making it harder for readers to immediately grasp how OFFSIDE concretely differs from prior datasets.***
>
>
>
> A1: Table 1 presents a comprehensive comparison between OFFSIDE and other related works. OFFSIDE is the first to support: (1) multi-image entity association (grouping images for each player), (2) selective unlearning of private attributes while preserving shared knowledge, (3) corrective relearning in a continual learning setting, and (4) unimodal unlearning (unlearning using only text data). Each of these settings introduces novel findings that have not been previously proposed.
>
> ------
> ***W2.1: How examples in the Relearn Set are derived from the Forget Set.***
>
>
> A2: Thanks for your kind reminder! We have provided a detailed explanation of how the relearn set is constructed in lines 192-196 (in red): $ D_{relearn}$ represents the corrected versions of the same rumors, offering new data about the same entities.
> Both the images and text in $ D_{relearn}$ are newly collected. This dataset is used to assess the effectiveness of unlearning methods in combination with post-training procedures, specifically evaluating the model's ability to recover knowledge that was previously unlearned through the relearning process.
>
> ------
> ***W2.2: Whether the Retain Set is revisited or frozen during relearning, and how the model’s stability on it is assessed.***
>
>
>
> A: During relearning, the Retain Set is frozen. The model's stability is assessed on the Retain Set, Test Set, MM-Bench (similar to the unlearning phase), and Relearn Set using the same metric. The relearning process is merely performed on the Relearn Set.
>
> ------
> ***W3: The temporal or procedural link between unlearning and relearning phases — i.e., whether the model is incrementally trained on new data or simply fine-tuned again.***
>
>
> A: The model is fine-tuned on relearn set(new data). This tests whether the unlearning method can work together with other post-training techniques.
>
> ------
> ***W4: The paper has limited novelty. Selective unlearning has been explored in previous multimodal unlearning datasets.***
>
> A: ***OFFSIDE's selective unlearning setting differs from prior work by enabling the targeted removal of specific image-text associations of a given image, preserving non-sensitive information, rather than indiscriminately deleting all associated data.***
> As stated in lines 80–81 of our paper: “The Selective Unlearning setting evaluates the ability to accurately erase specific image-text associations of a given image without affecting other benign knowledge.” In contrast, existing benchmarks [1,2,4] define “selective” unlearning as the removal of all private information associated with a given entity. However, OFFSIDE focuses on a more fine-grained scenario: for a single image that may be linked to multiple pieces of private information (e.g., 10 distinct facts), our benchmark requires unlearning only a subset of those associations (e.g., 4 out of 10 facts), while preserving the rest. This distinction is crucial because the retained information may be benign or publicly acceptable, and indiscriminately deleting all associated data would be unnecessarily destructive and the model might achieve unlearning by forgetting the entire entity. Our setting, therefore, better reflects real-world requirements where precision, rather than blanket removal.

---

> ### Author Response · Authors · 2025-11-18
> **Reply to questions**
>
> ------
> ***Q1: Why does the continual learning set be a part of the dataset? It can be any other dataset, right? Is there a reason for the way it is designed?***
>
> A: We include the relearning setting primarily to enable a more comprehensive evaluation of the robustness of existing baselines. Specifically, we aim to ensure that unlearning methods can seamlessly integrate with other post-training techniques to accomplish a complete downstream task. In the context of rumor mitigation, it is natural and indeed necessary to correct misinformation after selectively unlearning specific false claims. Therefore, we consider relearning an essential component of the pipeline.
>
> ------
> ***Q2: Is there a difference between the continual learning setting proposed in the paper and finetuning attacks proposed in previous papers?***
>
> A: The core novelty of our work lies in the introduction of **visual rumors**, a scenario that has not been explored in prior unlearning literature. Regarding why we choose continual learning set be a part of the dataset, please refer to Q1. While fine-tuning attacks, such as relearning[3], have been addressed in the LLM unlearning domains, this has not been explored in the context of MLLM unlearning. We include this aspect primarily to ensure a more comprehensive evaluation and to test whether the conclusions drawn from uni-modal models can be scaled to multi-modal scenarios.
>
>
> ------
>
> [1] Patil, Vaidehi, et al. "Unlearning Sensitive Information in Multimodal LLMs: Benchmark and Attack-Defense Evaluation." Transactions on Machine Learning Research.
>
> [2] Liu, Zheyuan, et al. "Protecting Privacy in Multimodal Large Language Models with MLLMU-Bench." Proceedings of the 2025 Conference of the Nations of the Americas Chapter of the Association for Computational Linguistics: Human Language Technologies (Volume 1: Long Papers). 2025.
>
> [3] Haoming Xu, et al. "ReLearn: Unlearning via Learning for Large Language Models". ACL 2025
>
> [4] AlexeyDontsov, et al. Clear: Character unlearning in textual and visual modalities. arXiv preprint arXiv:2410.18057,2024.

---

> ### Author Response · Authors · 2025-11-26
>
> We hope this message finds you well. As the discussion period is drawing to a close (with about one week remaining), we wanted to reach out to see if you might have any further comments, questions, or concerns that we could help address. We sincerely appreciate the time and thought you’ve invested in reviewing our work, and we would be more than happy to provide any additional information that could support your evaluation.
>
> Thank you once again for your kind attention and valuable feedback.

---

### Note · Authors · 2025-12-25

I have read and agree with the venue's withdrawal policy on behalf of myself and my co-authors.